# Effects of Dietary *Bacillus coagulans* and Tributyrin on Growth Performance, Serum Antioxidants, Intestinal Morphology, and Cecal Microbiota of Growing Yellow-Feathered Broilers

**DOI:** 10.3390/ani13223534

**Published:** 2023-11-15

**Authors:** Jinwang Hou, Lina Lian, Lizhi Lu, Tiantian Gu, Tao Zeng, Li Chen, Wenwu Xu, Guoqin Li, Hongzhi Wu, Yong Tian

**Affiliations:** 1State Key Laboratory for Managing Biotic and Chemical Threats to the Quality and Safety of Agro-Products, Institute of Animal Husbandry and Veterinary Medicine, Zhejiang Academy of Agricultural Sciences, Hangzhou 310021, China; hjw970214@163.com (J.H.); lianlinayx@163.com (L.L.); lulizhibox@163.com (L.L.); gtt19931029@126.com (T.G.); zengtao4009@126.com (T.Z.); chenli0429@163.com (L.C.); xuwenwu248@outlook.com (W.X.); ligq@zaas.ac.cn (G.L.); 2Tropical Crop Genetic Resource Research Institute, Chinese Academy of Tropical Agricultural Sciences, Haikou 571101, China

**Keywords:** *Bacillus coagulans*, tributyrin, serum antioxidant, gut microbes, yellow-feathered broiler

## Abstract

**Simple Summary:**

*Bacillus coagulans* and tributyrin are important feed additives and both can improve the health of animals when used alone. Here, we found that the combination of *Bacillus coagulans* and tributyrin glycerin improved the antioxidant capacity, intestinal morphology, and composition of the intestinal microbiota of yellow-feathered broilers to a certain extent. We further found that these effects were synergistic. Overall, our study provided insight into the development of an effective and safe blend of feed additives in the poultry industry.

**Abstract:**

This study investigated the impact of *Bacillus coagulans* (BC) and tributyrin (TB) supplementation on the growth performance, serum antioxidant capacity, intestinal morphology, and cecal microbiota of yellow-feathered broilers. Using a 2 × 2 factorial design, 480 broilers were randomly assigned to four experimental diets, comprising two levels of BC (0 and 1 g/kg) and two levels of TB (0 and 1 g/kg), over a 36-day period. A significant interaction was observed between BC and TB, impacting the average daily feed intake (ADFI) of broilers aged between 26 and 40 days (*p* < 0.01). BC and TB also displayed a significant interaction in relation to serum malondialdehyde levels and total antioxidant capacity (*p* < 0.05). Additionally, there was a significant interaction between BC and TB concerning the duodenal villus-to-crypt ratio, crypt depth, and jejunal villus-to-crypt ratio (*p* < 0.05). The addition of BC and TB significantly enhanced the richness and diversity of cecal microbiota, with a notable interactive effect observed for the abundance of Faecalibacterium, Ruminococcus_torques_group, and Phascolarctobacterium. In conclusion, supplementation with BC and TB can effectively improve the growth performance, serum antioxidant capacity, intestinal morphology, and cecal microbiota composition of yellow-feathered broilers, indicating the presence of an interactive effect.

## 1. Introduction

Poultry constitutes an increasingly pivotal protein source for global human consumption, underscoring its significance in the human diet [1]. The average daily gain (ADG) and average daily feed intake (ADFI) of commercial broilers have witnessed remarkable increments in recent years, while the breeding cycle of commercial broilers has undergone a shorter duration. This phenomenon may potentially augment the intestinal burden and foster the generation of reactive oxygen species within broilers, thereby inducing oxidative stress [2]. Enhancements in the growth performance of broiler chickens are intimately correlated with the amelioration of intestinal health [3]. The establishment of a well-balanced microbiome is indispensable for the development of a robust intestinal system [4]. Studies have demonstrated that the richness of gut microbiota mirrors the digestive and absorptive capabilities of the intestine, and a harmoniously balanced gut microbiome can facilitate the maturation and development of the host’s immune system, exerting a pivotal role in host health [5].

*Bacillus coagulans* is a probiotic Gram-positive bacterium characterized by its heightened stress resistance, as well as its ability to generate lactic acid and non-pathogenic spores. Accordingly, *Bacillus coagulans* is widely used as a probiotic additive in the diets of livestock and poultry, particularly in China [6,7,8,9,10,11]. Endres et al. [12] conducted a comprehensive toxicological safety assessment and chronic oral toxicity study on *Bacillus coagulans*, leading them to conclude that *Bacillus coagulans* is highly suitable as a feed additive [13]. In addition, spores generated by *Bacillus coagulans* exhibit remarkable stability and stress resistance, enabling them to withstand gastric acid and bile salts, thereby reaching the small intestine, where they germinate and proliferate. [14,15,16]. Studies have shown that *Bacillus coagulans* exhibits a remarkable capacity to enhance ADG, augment feed digestibility, and improve the composition of the intestinal microbiota of broilers [11]. *Bacillus coagulans* can secrete a variety of digestive enzymes, synthesize amino acids and vitamins, foster broiler growth, and enhance absorption efficiency during feed digestion [11,17]. *Bacillus coagulans* can also diminish intestinal pH, facilitate the proliferation of advantageous bacteria, and impede the growth of detrimental microorganisms [18]. The inclusion of *Bacillus coagulans* in the human diet can alleviate the damage caused by intestinal inflammation [19].

Butyrate is a short-chain fatty acid that plays a pivotal role in upholding the integrity of the intestinal mucosa [20,21], furnishing energy to intestinal mucosal cells, and fostering equilibrium within the gut microbiota [20,22]. Owing to the irritant properties and disagreeable odor of butyric acid, it is unsuitable for direct utilization in the production of livestock and poultry feed [23]; however, butyric acid derivatives, such as tributyrin, are frequently used in livestock and poultry production [24]. Tributyrin emerges from the esterification of one glycerol molecule and three butyrate molecules. It is very stable in the stomach and, upon entering the intestine, it slowly releases a large amount of butyrate through the activity of lipases [25,26,27]. This ability of tributyrin enables a greater amount of butyrate to reach the back of the small intestine [28]. A recent study demonstrated that the supplementation of tributyrin in the diets of piglets yielded a marked improvement in ADG and a significant enhancement in their intestinal morphology [29]. Furthermore, Yang et al. [30] elucidated that tributyrin supplementation can modulate the intestinal microflora of broilers.

We posit that the synergistic utilization of these components would yield greater efficacy in enhancing the intestinal microbiota environment of broiler chickens, promoting the healthy functioning of the gastrointestinal system, and fostering the overall well-being and growth of the animals. Consequently, we systematically assessed the effects of individually or jointly incorporating *Bacillus coagulans* and tributyrin into the diet on the growth performance, serum antioxidant capacity, intestinal morphology, and intestinal microbiota of yellow-feathered broiler chickens. This study bears considerable theoretical significance in advancing the intestinal health and growth performance of yellow-feathered broiler chickens.

## 2. Materials and Methods

### 2.1. Materials

*Bacillus coagulans* (live bacteria ≥ 2 × 10^8^ CFU/g) were obtained from Beijing Kangyuan Yisheng Biotechnology Co., Ltd. (Beijing, China); tributyrin (purity ≥ 64%) was purchased from Harbin Jinfulai Technology Development Co., Ltd. (Harbin, China).

### 2.2. Experimental Design and Animal Management

The yellow-feathered broiler chicken is a time-honored indigenous poultry breed in our country. With its tender meat and excellent flavor, it has gained wide popularity among consumers in China and neighboring nations. This study employed the intermediate-speed yellow-feathered broiler chicken breed. The growth cycle of this breed consists of a brooding period from 1 to 25 days old, and a growing period from 26 to 62 days old. In this study, we selected yellow-feathered broiler chickens at the age of 26 days as experimental subjects. During this stage, yellow-feathered broiler chickens exhibit rapid growth, excellent health condition, and vigorous metabolism.

A total of 480 yellow-feathered broiler chickens (26 days old, female birds, 370.52 ± 3.61 g) were used in this study. The experimental animals were kept in an identical rearing environment and conditions prior to their 26th day of age, ensuring complete homogeneity of treatment procedures at the commencement of the experiment. The experimental animals were randomly divided into four groups, each consisting of six replicates, with 20 individuals per replicate. The experiment followed a 2 × 2 factorial design, wherein two levels of *Bacillus coagulans* supplementation (0 and 1 g/kg) and two levels of tributyrin supplementation (0 and 1 g/kg) were incorporated into the basal experimental diet. The duration of the trial was 36 days (the individual dosages of *Bacillus coagulans* and tributyrin were determined based on the optimal dosage provided by the reagent company). The CON group was fed the basal experimental diet, the BC group had 1 g/kg *Bacillus coagulans* added to the basal experimental diet, the TB group had 1 g/kg tributyrin added to the basal experimental diet, and the BC*TB group had 1 g/kg *Bacillus coagulans* and 1 g/kg tributyrin added to the basal experimental diet in combination. The trial was conducted at the experimental site of Jiangxi Qiling Agriculture and Animal Husbandry Company. Before the trial, birds in each replicate were maintained in separate pens and each pen was thoroughly cleaned and disinfected. All enclosures were maintained under identical feeding conditions. Rice husks were evenly spread on the ground (4–5 cm) of the chicken pens and the animals had free access to clean water and feed. Continuous lighting was provided 24 h per day. The study was approved by the Laboratory Animal Ethics Committee of the Zhejiang Academy of Agricultural Sciences (Hangzhou, China). The basal diet was formulated based on the Agricultural Industry Standard of the People’s Republic of China—Chicken Feed (NY_T33-2004) [31] and modified in accordance with production practices. The dietary and nutritive composition of the provided feed is shown in Table 1 (pelleted).

### 2.3. Sample Collection

At 62 days of age, following a 12 h period of fasting, one experimental broiler chicken, approximately matching the average body weight, was selected from each replicate (comprising 6 birds per group) for sample collection. Blood samples were collected via the wing vein and then centrifuged at 3000 rpm for 10 min at 4 °C for the determination of serum antioxidant indexes. Additionally, 2–3 cm of tissue was collected from the duodenum, jejunum, and ileum of each bird and fixed in 4% paraformaldehyde for analysis of mucosal morphology. Finally, the cecal contents of the animals were collected, placed in 2 mL cryovials, and stored in liquid nitrogen for microbial 16S rRNA gene sequencing.

### 2.4. Growth Performance

Feed intake was recorded daily for the determination of the average daily feed intake (ADFI), which was calculated as follows: ADFI = (feed intake during the test−feed remaining during the test)/number of test days. The weight of each group of broiler chickens was measured at 26, 40, and 62 days of age for the determination of the average daily gain (ADG), which was calculated as follows: ADG = (test final weight-test initial weight)/number of test days. The feed-to-gain ratio (F/G) was calculated as ADFI/ADG.

### 2.5. Measurement of Serum Antioxidant Indexes

After thawing the serum, the corresponding reagent kits (provided by the HuaYing Institute of Biotechnology, Beijing, China) were employed to determine the total antioxidant capacity (T-AOC) (T-AOC (HY-60021) KIT; minimal detection = 50 μL), malondialdehyde content (MDA) (MDA (HY-M0003) KIT; minimal detection = 50 μL), as well as the activities of catalase (CAT) (CAT (HY-M0018) KIT; minimal detection = 50 μL), superoxide dismutase (SOD) (SOD (HY-M0001) KIT; minimal detection = 50 μL), and glutathione peroxidase (GSH-PX) (GSH-PX (HY-M0004) KIT; minimal detection = 50 μL) in the serum, following the instructions provided by the manufacturers.

### 2.6. Intestinal Morphology

Samples of 2 cm were taken from various parts of the small intestine (duodenum, jejunum, ileum) at a quantity of 6 per group. These samples were fixed in a 4% formaldehyde solution for 72 h, followed by dehydration, cleaning, embedding in wax, sectioning, and staining with hematoxylin and eosin. To determine the height of villi and the depth of crypts, a photomicroscope (Eclipse Ci-L, Nikon, Japan) was used to capture images at 40× magnification. The background lighting of each photograph was kept consistent during imaging, and analysis was performed using ImagePro Plus 6.0 software (Media Cybernetics, Bethesda, MD, USA). The height of the villi was measured from the tip (including the lamina propria) to the base (the junction between villi and crypts), while the depth of the crypts was calculated from the junction between villi and crypts to the far end border of the crypts. The villi-height-to-crypt-depth (V/C) ratio was then calculated based on the aforementioned parameters.

### 2.7. 16S rRNA Sequencing

16S rRNA sequencing of the cecal intestinal contents of 24 animals (six replicates per group, one sample per replicate) was conducted, and differences in the cecal intestinal microflora among the four groups were characterized. Total DNA was extracted from cecal intestinal contents using the QIAamp DNA Stool Mini Kit (QIAGEN, CA, Hamburg, Germany) following the manufacturer’s protocol. The average DNA purity (A230/A260) was 2.3 and the average DNA concentration was 100 µg/mL. The V3–V4 region of the 16S rRNA gene was amplified by PCR using primers 341F (5′-CCTAYGGRBGCASCAG-3′) and 806R (5′-GGACTACNNGGGTATCTAAT-3′). The enzyme and buffer solution employed was Phusion@ High-Fidelity PCR Master Mix with GC Buffer (New England Biolabs, Ipswich, MA, USA). All PCR reactions were carried out in 30 μL reactions with 15 μL of Phusion@High-Fidelity PCR Master Mix (New England Biolabs), 0.2 μM of forward and reverse primers, and about 10 ng template DNA. Thermal cycling consisted of initial denaturation at 98 °C for 1 min, followed by 30 cycles of denaturation at 98 °C for 10 s, annealing at 50 °C for 30 s, elongation at 72 °C for 60 s, and finally 72 °C for 5 min. The purity of the PCR products was assessed by 2% agarose gel electrophoresis and the products were purified using the GeneJET DNA Gel Extraction Kit (Thermo Fisher Scientific, Waltham, MA, USA). A DNA library was established using the TruSeq DNA PCR-Free Sample Preparation Kit, quantified and verified by Qubit, and sequenced using the Illumina Novaseq 6000 platform (Illumina, San Diego, CA, USA). The raw reads were demultiplexed, quality-filtered by Trimmomatic, and reads were assembled using FLASH software (v.1.2.7). All samples were clustered into operational taxonomic units (OTUs) with Uparse software (Uparse v7.0.1001) at a 97% similarity threshold [32]. Species annotation was performed on the sequences within the OTUs according to the Greengenes database (http://greengenes.secondgenome.com/ (accessed on 21 December 2021)).

### 2.8. Statistical Analysis

A 2 × 2 factorial analysis of variance (two-way ANOVA) was conducted using the GLM procedure in SPSS 26.0 (IBM, Chicago, IL, USA). The statistical model included the factors of adding *Bacillus coagulans*, adding tributyrin, and their interaction. Data were expressed as means ± SEM and the threshold for statistical significance was set at *p* < 0.05. Alpha diversity was estimated by the Chao1, Ace, Shannon, and Simpson indices. Beta diversity assessment was based on the weighted UniFrac distance matrix and visualized by principal coordinates analysis (PCoA). In the LEfSe analysis, the non-parametric Kruskal–Wallis rank-sum test was used to detect species displaying significantly differential abundance between groups, followed by the paired Wilcoxon rank-sum test. Linear discriminant analysis (LDA) was used to estimate the effect size of each differentially abundant species.

## 3. Results

### 3.1. Growth Performance

The growth performance of yellow-feathered broiler chickens is shown in Table 2. Supplementation with *Bacillus coagulans* decreased the ADFI from 26 to 40 days of age (*p* < 0.01) and the F/G ratio from 26 to 62 and from 41 to 62 days of age (both *p* < 0.01). *Bacillus coagulans* supplementation also increased the ADG between the ages of 26 and 62 days and between the ages of 41 and 62 days (*p* < 0.01). Meanwhile, tributyrin supplementation decreased the ADFI from 26 to 40 days of age (*p* < 0.01). The combined supplementation of *Bacillus coagulans* and tributyrin decreased the ADFI between the ages of 26 and 40 days (*p* < 0.01).

### 3.2. Serum Antioxidant Indexes

The serum antioxidant indexes of yellow-feathered broiler chickens are shown in Table 3. *Bacillus coagulans* supplementation increased the activities of SOD, GSH-Px, and CAT (*p* < 0.05), while tributyrin supplementation increased the activity of GSH-Px and decreased the content of MDA (*p* < 0.05). Combined *Bacillus coagulans* and tributyrin supplementation decreased MDA content and increased T-AOC (*p* < 0.05).

### 3.3. Intestinal Morphology

Villus height, crypt depth, and V/C ratio in the intestines of broiler chickens are shown in Table 4 and Figure 1. The addition of *Bacillus coagulans* to the feed significantly decreased crypt depth in the duodenum, jejunum, and ileum (*p* < 0.05) and increased the V/C ratio in the duodenum and jejunum (*p* < 0.05). The addition of tributyrin to the feed significantly increased villus height in the duodenum and jejunum (*p* < 0.05), decreased crypt depth in the duodenum and ileum (*p* < 0.01), and increased the V/C ratio in the duodenum and ileum (*p* < 0.05). The combined supplementation of *Bacillus coagulans* and tributyrin significantly affected duodenal crypt depth and V/C ratio as well as ileal V/C ratio (*p* < 0.05).

### 3.4. Diversity, Richness, and Composition of the Bacterial Communities in the Cecum 

A total of 2 379 916 effective high-quality sequences were screened out. To investigate the species composition of the samples, clean reads were clustered into OTUs according to a 97% similarity threshold. Each OTU corresponded to one microbial species, with 2 482 OTUs being identified in total. The number of unique OTUs in each group was 191, 221, 271, and 300 (Figure 2). Alpha diversity is shown in Table 5. Tributyrin supplementation significantly increased the observed species and ACE index values (*p* < 0.05); the Chao 1 index was also higher in this group, although not significantly (*p* = 0.059). The addition of *Bacillus coagulans* or a combination of *Bacillus coagulans* and tributyrin to the diet of broiler did not significantly affect alpha index values (both *p* > 0.05). Beta diversity, visualized by PCoA, is depicted in Figure 3. PcoA1 and PcoA2 explained 11.68% and 8.049% of the variation in the data, respectively; no significant differences in microbial beta diversity were observed among the groups. The relative microbial abundances at the phylum and genus levels were also assessed. At the phylum level, *Firmicutes*, *Bacteroidetes*, *Proteobacteria*, *Actinobacteria*, *Epsilonbacteraeota*, *Deferribacteres*, *Synergistetes*, *Tenericutes*, Cyanobacteria, and *Elusimicrobia* were the 10 most abundant bacteria in the guts of yellow-feathered broiler chickens. Firmicutes and Bacteroidetes were the dominant phyla in the cecal microbiota of each group (Figure 4). The addition of *Bacillus coagulans* alone increased the abundance of *Tenericutes* (*p* < 0.05) but had no significant effect on other phyla (*p* > 0.05). The addition of tributyrin significantly reduced the abundance of *Epsilonbacteraeota* (*p* < 0.05) but had no significant effect on other phyla (*p* > 0.05). The combination of the two additives had a significant interaction on Firmicutes abundance (*p* < 0.05) (Table 6). The 10 most abundant genera accounted for approximately 70% of the cecal microbiota. At the genus level, *Bacteroides*, *Faecalibacterium*, *Rikenellaceae RC9 gut group*, *[Ruminococcus] torques group*, *Ruminococcaceae UCG-014*, *Desulfovibrio*, *Phascolarctobacterium*, *Megamonas*, *Alistipes*, and *Mucispirillum* were the dominant cecal microbiota in all the groups (Figure 5). As shown in Table 6, *Bacillus coagulans* or tributyrin supplementation alone did not significantly affect bacterial abundance at the genus level (*p* > 0.05); in combination, however, *Bacillus coagulans* and tributyrin supplementation had a significant interaction on *Faecalibacterium*, *[Ruminococcus] torques group*, and *Phascolarctobacterium* abundance (*p* < 0.05).

We also characterized the differences in bacterial abundance among the *Bacillus coagulans* (BC), tributyrin (TB), and BC × TB groups relative to the control group using LefSe analysis. The results revealed that 28 bacterial clades at all taxonomic levels were differentially abundant (LDA scores >2.0 and *p* < 0.05) (Figure 6). The abundances of *o_Selenomonadale*, *c_Negativicutes*, *f_Acidaminococcaceae*, *g_Phascolarctobacterium*, *g_Campylobacter*, *f_Campylobacteraceae*, *c_Campylobacteria*, *o_Campylobacterales*, *p_Epsilonbacteraeota*, and *g_Candidatus_Saccharimonas* were highest in the control group. The abundances of *g_Marvinbryantia*, *c_Spirochaetia*, *o_Frankiales*, *f_Sporichthyaceae*, and *g_Sphaerochaeta* were highest in the BC group. The abundance of *g_Flavobacterium* was highest in the TB group. Finally, the abundances of *f_Ruminococcaceae*, *f_Barnesiellaceae*, *g_uncultured_rumen_bacterium*, *f_Streptococcaceae*, *g_Allobaculum*, and *g_Providencia* were highest in the BC × TB group.

## 4. Discussion

Growth performance is an important indicator of the efficiency of livestock and poultry production. The inclusion of *Bacillus coagulans* or tributyrin in the diet has been shown to enhance the growth performance of livestock and poultry [9,33]. Research has indicated that the inclusion of *Bacillus coagulans* in the diet of Guangxi yellow-feathered broilers improves their growth performance and significantly reduces the F/G ratio in broilers aged 21 to 42 days [34]. Another study showed that dietary supplementation with *Bacillus coagulans* can augment the ADG and ADFI of yellow-feathered broiler chickens [11]. Consistent with these results, our investigation revealed that the addition of *Bacillus coagulans* to the diet significantly increased the ADG from 41 to 62 and from 26 to 62 days of age, while concurrently reducing the F/G ratio from 26 to 62 and from 41 to 62 days of age. The reduction in feed intake might be related to the nutrient utilization of broilers [35], which needs further verification. Dietary supplementation with tributyrin has been shown to improve animal production performance [27,29,36]. However, the efficacy of tributyrin as a growth promoter remains controversial. Bedford et al. [37] found that the addition of tributyrin did not significantly impact the ADG or feed conversion rate of broilers. In our study, the addition of tributyrin significantly decreased the ADFI of chickens aged 26 to 40 days, while also improving other growth performance indicators, albeit not significantly. These differences in the effects of tributyrin might stem from several factors, including differences in the level of supplementation, as well as the breed and age of the broilers used. Combined supplementation with *Bacillus coagulans* and tributyrin exerted more substantial effects than the addition of the individual supplements, including an increase in the ADFI between the ages of 26 and 40 days. The mechanism underlying this effect necessitates further study.

Redox status can reflect the health of animals to a large extent, and both livestock and poultry express a variety of antioxidant enzymes that act to regulate the redox balance [38]. The prolonged accumulation of peroxides can induce oxidative damage to lipids, proteins, and nucleic acids [39]. Lipid oxidation augments the conversion of myoglobin to oxymyoglobin, consequently exerting deleterious effects on the palatability and nutritional composition of chicken meat [40]. *Bacillus coagulans* can enhance the levels of SOD and CAT, while reducing the levels of the lipid peroxidation biomarker MDA, thereby mitigating the deleterious impact of oxidative species [41]. Zhang et al. [42] found that *Bacillus coagulans* supplementation can significantly increase GSH-Px, SOD, and CAT activities and markedly reduce MDA content in the serum of broilers. Similarly, we found that *Bacillus coagulans* supplementation increased SOD, GSH-Px, and CAT activities in the serum of yellow-feathered broiler chickens. Meanwhile, Wang et al. [27] reported that the addition of tributyrin to the diet of 48-week-old broiler breeders significantly increased the T-AOC and substantially decreased the MDA content. In agreement with these studies, we found that tributyrin supplementation reduced MDA concentrations and increased the activity of GSH-Px. Xing et al. [43] demonstrated that incubation with sodium butyrate significantly upregulated the expression of the transcription factor Nrf2 in oxidation-damaged cells and enhanced their antioxidant capacity. Nrf2 has important antioxidant activity, regulating the expression of a variety of antioxidant enzyme-coding genes [44]. These observations suggest that the effect of the combined supplementation of *Bacillus coagulans* and tributyrin on T-AOC and MDA concentrations observed in this study may be due to the combined antioxidant effects of the two additives.

In addition to its secretion of digestive fluids, the small intestine serves as the primary site for the digestion and absorption of nutrients [45]. Maintaining the integrity of the intestinal barrier is a prerequisite for the healthy growth of animals and their ability to withstand chemical and microbial threats [46]. In our study, both *Bacillus coagulans* and tributyrin exerted beneficial effects on the intestinal morphology of yellow-feathered broiler chickens. Villus length, crypt depth, and V/C ratio are important indicators for assessing intestinal maturity and functionality [47]. Crypt depth reflects epithelial regenerative speed, with shallower crypts indicating faster cell renewal and increased secretion activity [48], but deeper crypts lead to more basal cell production, resulting in decreased mature epithelial cells and reduced nutrient utilization efficiency [49]. The higher the villus height, the greater the V/C ratio, indicating a stronger capacity for digestion and absorption [50]. According to reports, the supplementation of *Bacillus coagulans* in piglets has been shown to significantly enhance the V/C ratio within the small intestine [9,51]. Similarly, the addition of tributyrin has been found to effectively increase both villus height and the V/C ratio in the intestinal tract of piglets [52]. This might stem from the entry of *Bacillus coagulans* into the intestine, which promotes the proliferation of beneficial bacteria, reduces the population of harmful bacteria, elevates the activity of digestive enzymes in the intestines (as a result of metabolites produced by beneficial bacteria), promotes the full decomposition of tributyrin, supplying greater energy to intestinal cells (through the decomposition of butyric acid), and accelerates the regeneration of intestinal epithelial cells.

Multiple studies have indicated that the gut microbiota is a crucial determinant of the host’s health status, playing a pivotal role in nutrient absorption, protection against harmful bacteria, improvement of growth and metabolism, as well as regulation of the immune system [53,54,55]. In this study, we employed 16S rRNA gene sequencing technology to analyze the characteristics of the microbial communities in the cecal contents of broiler chickens. Compared to the control group, the dietary supplementation of *Bacillus coagulans* or tributyrin showed a certain degree of enhancement in the richness and diversity of the cecal microbial community in broiler chickens, although the ACE, Chao 1, Shannon, and observed species indexes did not exhibit a significant increase. Firmicutes is the most common microbial phylum in the intestinal environment of many birds [56,57,58]. Members of this phylum are associated with polysaccharide degradation, butyrate production, and enhanced nutrient absorption capacity [59,60]. Zhang et al. [52] found that the relative abundance of Firmicutes in weaned piglets was increased under supplementation with tributyrin and essential oils. This effect may be explained by the fact that tributyrin decomposition led to the release of a large amount of butyrate in the intestine, which reduced intestinal pH, thus improving the status of the intestinal flora. In our study, combined *Bacillus coagulans* and tributyrin supplementation increased the relative abundance of Firmicutes to a greater extent than either additive alone, indicative of a synergistic effect between the two additives. It has been shown that the occurrence of obesity in mice is positively correlated with changes in the abundance of *Tenericutes* [61]. In our investigation, it was observed that all three experimental groups exhibited higher ADG in comparison to the control group. This correlation could potentially be linked to the observed increase in the abundance of *Tenericutes*; however, further validation is required to substantiate this hypothesis. Meanwhile, antibiotic supplementation was reported to significantly reduce the relative abundance of *Epsilonbacteraeota* [62], which was in line with our findings. In our study, the addition of *Bacillus coagulans* and tributyrin reduced the abundance of *Epsilonbacteraeota*, with the best effect being observed with the combined addition of *Bacillus coagulans* and tributyrin.

A significant interaction effect was observed between the coadministration of *Bacillus coagulans* and tributyrin, specifically impacting the abundances of *Faecalibacterium*, *Phascolarctobacterium*, and the *[Ruminococcus] torques group*. The combined application of *Bacillus coagulans* and tributyrin yielded greater increases in the abundances of *Faecalibacterium* and *Phascolarctobacterium* compared to the supplementation of either additive alone. These bacterial species are crucial for the production of short-chain fatty acids [63], with *Faecalibacterium* additionally capable of secreting anti-inflammatory substances, thereby reducing intestinal inflammation [64,65]. Notably, Sun et al. [66] demonstrated a significant influence of *Bacillus coagulans* supplementation on *Faecalibacterium* abundance in weaned piglets’ intestines. However, in our study, a similar effect was observed for the combination of *Bacillus coagulans* and tributyrin, but not for *Bacillus coagulans* supplementation alone. Disparities in *Bacillus coagulans* concentrations or the interplay between *Bacillus coagulans* and tributyrin might account for these divergent outcomes. *Phascolarctobacterium*, a bacterium responsible for propionate production, exhibits reduced abundance in cases of colonic inflammation [67,68]. Gong et al. [69] demonstrated that tributyrin supplementation elevates *Phascolarctobacterium* concentrations in yellow-feathered broiler chickens. Our study revealed higher *Phascolarctobacterium* abundance in animals supplemented with *Bacillus coagulans* and tributyrin, highlighting a significant interaction effect between the two additives. Overall, our findings indicate that the supplementation of *Bacillus coagulans* and tributyrin stimulates the growth of beneficial intestinal microorganisms while inhibiting the proliferation of pathogenic bacteria.

## 5. Conclusions

Our findings demonstrate that the inclusion of *Bacillus coagulans* (2 × 10^8^ CFU/g) at a dosage of 1 g/kg, along with tributyrin (purity ≥ 64%) at the same dosage, not only enhanced the antioxidant capacity but also ameliorated intestinal morphology, while simultaneously augmenting the richness and diversity of microflora in the cecum of broilers. Additionally, the supplementation of *Bacillus coagulans* alone yielded a remarkable improvement in the growth performance of yellow-feathered broilers. Notably, the combined administration of *Bacillus coagulans* and tributyrin exhibited a substantial interaction effect on growth performance, serum antioxidant capacity, intestinal morphology, and microbial richness and diversity in the cecum of yellow-feathered broilers. These findings expand our comprehension of the positive synergistic effects of *Bacillus coagulans* and tributyrin on the growth of yellow-feathered broilers. Furthermore, our results provide valuable insights for the development of environmentally friendly, safe, and pollutant-free feed additives.

## Figures and Tables

**Figure 1 animals-13-03534-f001:**
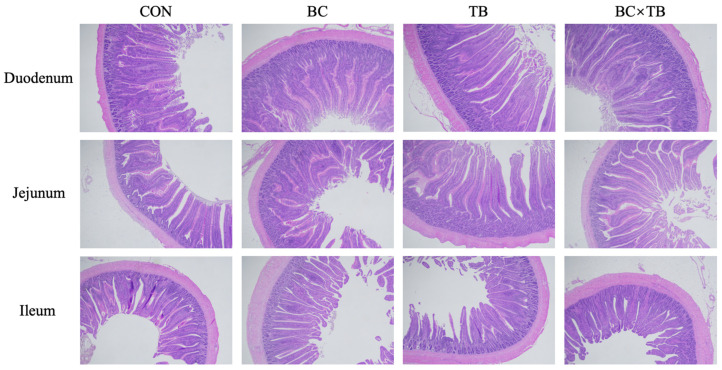
Photomicrograph of the small intestine (scale bar = 500 μm). Abbreviations: CON, control group (fed a basal diet); BC, the group fed a basal diet supplemented with 1 g/kg *Bacillus coagulans*; TB, the group fed a basal diet supplemented with 1 g/kg tributyrin; BC × TB, the group fed a basal diet supplemented with 1 g/kg *Bacillus coagulans* and 1 g/kg tributyrin.

**Figure 2 animals-13-03534-f002:**
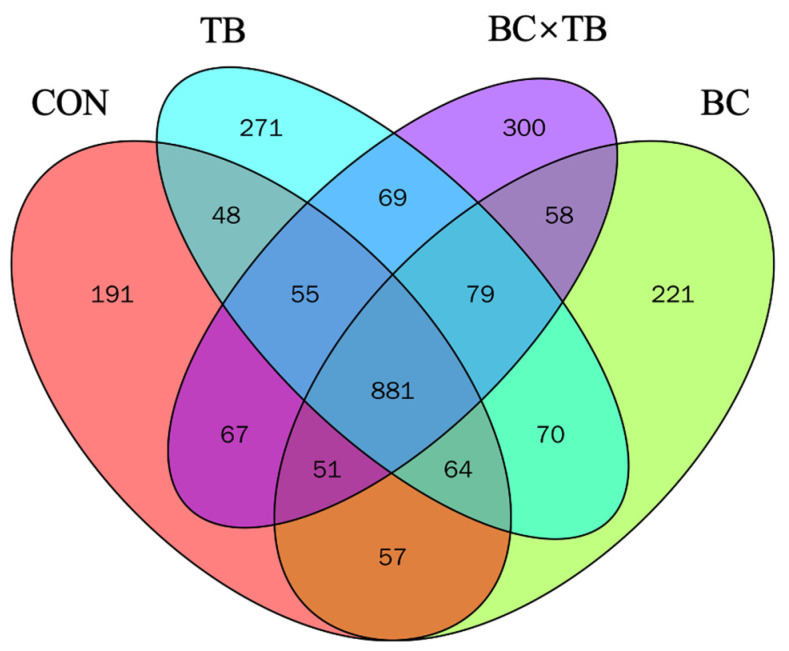
Analysis of the operational taxonomic units (OTUs) shared among groups. Each circle in the figure indicates a group, and numbers in overlapping circles indicate the number of OTUs shared between groups. Numbers located in non-overlapping areas indicate the number of unique OTUs in each group. Abbreviations: CON, control group (fed a basal diet); BC, the group fed a basal diet supplemented with 1 g/kg *Bacillus coagulans*; TB, the group fed a basal diet supplemented with 1 g/kg tributyrin; BC × TB, the group fed a basal diet supplemented with 1 g/kg *Bacillus coagulans* and 1 g/kg tributyrin.

**Figure 3 animals-13-03534-f003:**
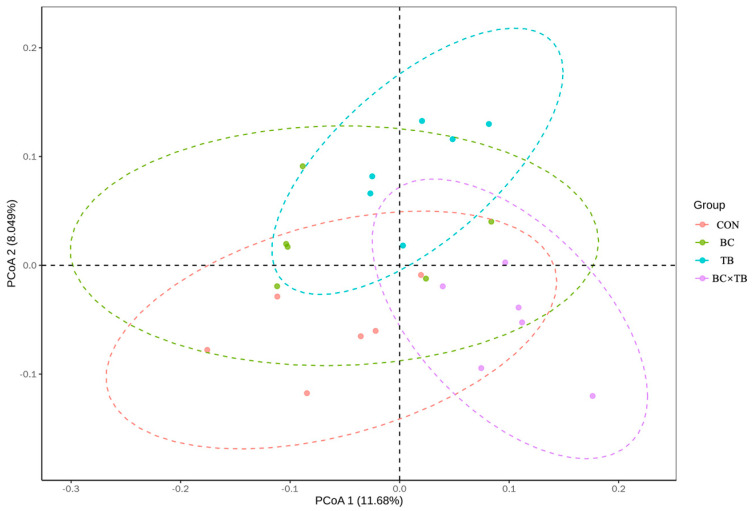
Principal coordinate analysis (PCoA) of cecal microbiota based on weighted UniFrac distances (*n* = 6). PCoA1, first principal coordinate; PCoA2, second principal coordinate. Abbreviations: CON, control group (fed a basal diet); BC, the group fed a basal diet supplemented with 1 g/kg *Bacillus coagulans*; TB, the group fed a basal diet supplemented with 1 g/kg tributyrin; BC × TB, the group fed a basal diet supplemented with 1 g/kg *Bacillus coagulans* and 1 g/kg tributyrin.

**Figure 4 animals-13-03534-f004:**
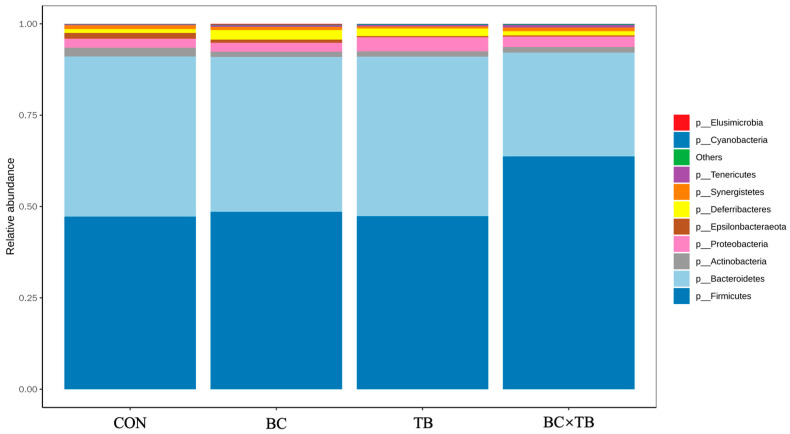
Histogram of the abundances of microbial phyla in the cecum. Abbreviations: CON, control group (fed a basal diet); BC, the group fed a basal diet supplemented with 1 g/kg *Bacillus coagulans*; TB, the group fed a basal diet supplemented with 1 g/kg tributyrin; BC × TB, the group fed a basal diet supplemented with 1 g/kg *Bacillus coagulans* and 1 g/kg tributyrin.

**Figure 5 animals-13-03534-f005:**
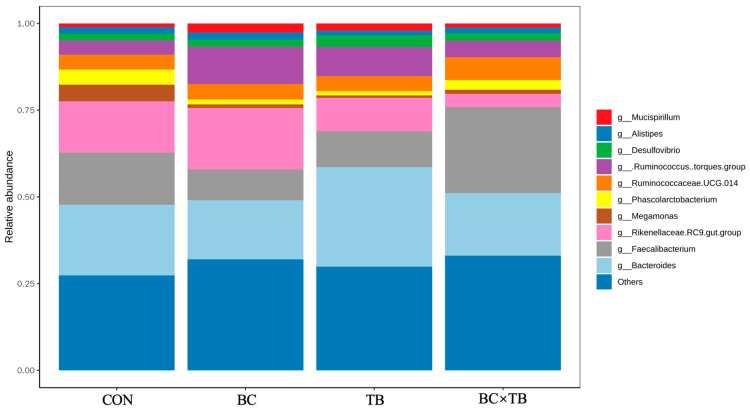
A histogram of the relative abundances of microbial genera in the cecum. Abbreviations: CON, control group (fed a basal diet); BC, the group fed a basal diet supplemented with 1 g/kg *Bacillus coagulans*; TB, the group fed a basal diet supplemented with 1 g/kg tributyrin; BC × TB, the group fed a basal diet supplemented with 1 g/kg *Bacillus coagulans* and 1 g/kg tributyrin.

**Figure 6 animals-13-03534-f006:**
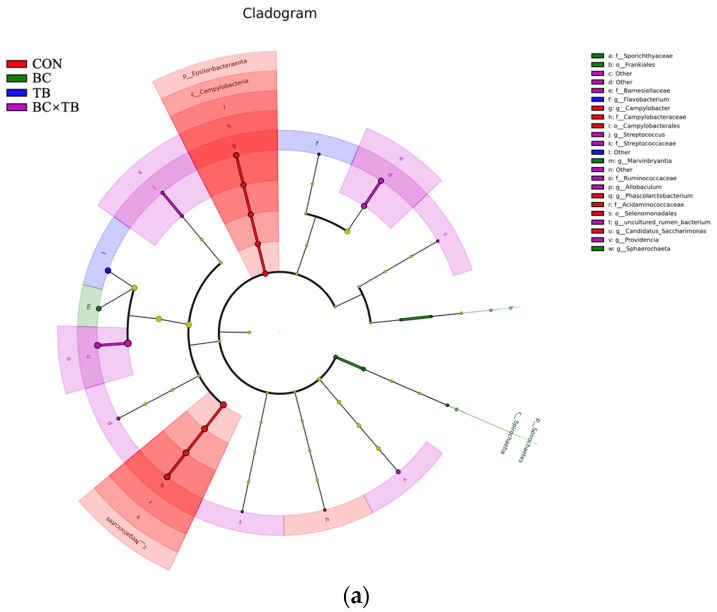
Linear discriminant analysis (LDA) effect size (LEfSe) was used to identify the most differentially abundant taxa in the cecal microbiota of the CON, BC, TB, and BC × TB groups. (**a**) A cladogram generated from LEfSe analysis. Red circles indicate taxa that were more abundant in the CON group, green circles indicate taxa that were more abundant in the BC group, blue circles indicate taxa that were more abundant in the TB group, purple circles indicate taxa that were more abundant in the BC × TB group, and yellow indicates taxa that were not differentially abundant. The diameters of the circles are proportional to each taxon’s abundance. (**b**) A histogram of the LDA scores computed for differentially abundant taxa in the BC group (green bars), TB group (blue bars), BC × TB group (purple bars), and CON group (red bars). Species displaying significant differences in abundance with an LDA score greater than 2.0 are shown. The length of the histogram indicates the LDA score, which can be interpreted as the effect size of each differentially abundant taxon. Abbreviations: CON, control group (fed a basal diet); BC, the group fed a basal diet supplemented with 1 g/kg *Bacillus coagulans*; TB, the group fed a basal diet supplemented with 1 g/kg tributyrin; BC × TB, the group fed a basal diet supplemented with 1 g/kg *Bacillus coagulans* and 1 g/kg tributyrin.

**Table 1 animals-13-03534-t001:** Composition and nutrient levels of the basal diet (as dry basis, %).

Ingredients, %	Content		Nutrient Levels	Content	
	26 to 40 Days	41 to 62 Days		26 to 40 Days	41 to 62 Days
Wheat	77.96	81.51	ME ^3^, MJ/kg	12.76	12.97
Soybean meal 43%	5.65	0.00	Crude protein, %	17.50	16.50
Sunflower meal	3.50	4.50	Crude fat, %	4.76	5.17
Peanut meal	4.00	4.25	Crude fiber, %	2.39	2.31
Corn gluten meal 60%	1.56	1.28	Calcium, %	0.80	0.75
Feather meal	0.00	1.00	Sodium, %	0.19	0.20
CaHPO4	0.63	0.47	Available phosphorus, %	0.30	0.28
Limestone	1.36	1.33	Lysine, %	0.90	0.85
Soybean oil	3.14	3.46	Methionine, %	0.45	0.37
Premix	2.20 ^1^	2.20 ^2^			
Total	100.00	100.00			

^1^ Premix provided per kilogram of diet from 26 to 40 days of age consisted of the following: NaHCO_3_ 2 g, NaCl 2 g, methionine 2.17 g, threonine 1.73 g, lysine sulphate 7.95 g, VA 10,000 IU, VD3 3,000 IU, VE 30 mg, VK3 1.3 mg, VB6 4 mg, VB12 0.013 mg, thiamine 2.2 mg, riboflavin 8 mg, nicotinamide 40 mg, choline chloride 600 mg, calcium pantothenate 10 mg, biotin 0.04 mg, folic acid 1 mg, Fe 80 mg, Cu 7.5 mg, Mn 110 mg, Zn 65 mg, I 1.1 mg, Se 0.3 mg. ^2^ Premix provided per kilogram of diet from 41 to 62 days of age consisted of the following: methionine 1.50 g, threonine 1.97 g, lysine sulphate 8.93 g. The rest of the ingredients was the same as those of the 26- to 40-day-old premix. ^3^ ME = metabolizable energy.

**Table 2 animals-13-03534-t002:** Effect of dietary supplementation with BC and TB on the growth performance of yellow-feathered broiler chickens (*n* = 6).

Items ^1^	Treatment				SEM	*p*-Value		
	TB, 0 g/kg		TB, 1 g/kg					
	BC, 0 g/kg	BC, 1 g/kg	BC, 0 g/kg	BC, 1 g/kg		BC	TB	BC × TB
26 to 40 days								
ADFI, g	57.23	54.64	53.88	54.14	0.29	0.001	0.001	0.001
ADG, g	23.28	23.24	23.21	23.10	0.14	0.846	0.788	0.910
F/G, g/g	2.46	2.35	2.32	2.35	0.02	0.338	0.104	0.120
41 to 62 days								
ADFI, g	70.02	69.83	69.75	70.17	0.11	0.605	0.891	0.172
ADG, g	18.88	22.58	19.13	21.25	0.42	0.001	0.329	0.241
F/G, g/g	3.72	3.11	3.68	3.31	0.07	0.001	0.387	0.323
26 to 62 days								
ADFI, g	64.76	64.43	64.06	64.45	0.09	0.853	0.061	0.084
ADG, g	20.69	22.86	20.81	22.01	0.25	0.001	0.296	0.243
F/G, g/g	3.13	2.83	3.09	2.93	0.04	0.001	0.546	0.212

^1^ ADFI = average daily feed intake; ADG = average daily gain; F/G ratio = feed-to-gain ratio.

**Table 3 animals-13-03534-t003:** Effect of dietary supplementation with BC and TB on the serum antioxidant index of yellow-feathered broiler chickens (*n* = 6).

Items ^1^	Treatment				SEM	*p*-Value		
	TB, 0 g/kg		TB, 1 g/kg					
	BC, 0 g/kg	BC, 1 g/kg	BC, 0 g/kg	BC, 1 g/kg		BC	TB	BC × TB
SOD, U/mL	57.60	72.03	62.89	75.47	2.37	0.003	0.284	0.818
GSH-Px, U/mL	137.29	162.46	161.35	168.19	3.66	0.012	0.018	0.128
MDA, nmol/mL	4.34	4.00	3.88	3.99	0.05	0.204	0.016	0.019
CAT, U/mL	47.67	54.41	51.78	60.11	1.46	0.005	0.053	0.743
T-AOC, U/mL	7.65	9.19	8.56	7.99	0.24	0.299	0.750	0.031

^1^ T-AOC = total antioxidant capacity; CAT = catalase; GSH-Px = glutathione peroxidase; SOD = superoxide dismutase; MDA = malonaldehyde.

**Table 4 animals-13-03534-t004:** Effect of dietary supplementation with BC and TB on the intestinal morphology of yellow-feathered broiler chickens (*n* = 6).

Items	Treatment				SEM	*p*-Value		
	TB, 0 g/kg		TB, 1 g/kg					
	BC, 0 g/kg	BC, 1 g/kg	BC, 0 g/kg	BC, 1 g/kg		BC	TB	BC × TB
Duodenum								
Villus height, µm	1202.14	1201.24	1244.58	1325.01	15.22	0.108	0.002	0.101
Crypt depth, µm	210.63	138.39	133.50	139.59	9.14	0.023	0.010	0.009
V/C ^1^	6.07	8.73	9.42	9.31	0.36	0.008	0.001	0.005
Jejunum								
Villus height, µm	1019.24	1079.39	1229.79	1273.36	27.10	0.123	0.001	0.799
Crypt depth, µm	153.88	73.92	136.77	101.39	8.61	0.001	0.676	0.083
V/C ^1^	7.36	14.88	10.33	13.43	0.80	0.001	0.942	0.335
Ileum								
Villus height, µm	609.13	603.32	622.38	675.85	14.88	0.431	0.164	0.329
Crypt depth, µm	142.29	129.73	125.63	88.90	6.16	0.020	0.008	0.230
V/C ^1^	4.73	4.68	5.10	7.76	0.40	0.051	0.013	0.044

^1^ V/C: villus height/crypt depth.

**Table 5 animals-13-03534-t005:** Effects BC and TB addition on alpha diversity of cecal microflora of yellow-feathered broiler chickens.

Items	Treatment				SEM	*p*-Value		
	TB, 0 g/kg		TB, 1 g/kg					
	BC, 0 g/kg	BC, 1 g/kg	BC, 0 g/kg	BC, 1 g/kg		BC	TB	BC × TB
Observed species	702.67	739.50	783.83	828.67	20.36	0.298	0.037	0.918
Shannon	5.28	5.42	5.54	5.68	0.14	0.638	0.402	0.993
Simpson	0.91	0.90	0.90	0.91	0.01	0.992	0.891	0.847
Chao1	808.10	831.80	869.75	924.31	20.14	0.321	0.059	0.693
ACE	826.93	850.80	905.02	949.26	21.96	0.427	0.049	0.811

**Table 6 animals-13-03534-t006:** Effects of BC and TB supplementation on microbial phyla and genera in the cecum of yellow-feathered broiler chickens.

Items	Treatment				SEM	*p*-Value		
	TB, 0 g/kg		TB, 1 g/kg					
	BC, 0 g/kg	BC, 1 g/kg	BC, 0 g/kg	BC, 1 g/kg		BC	TB	BC × TB
Phyla								
Firmicutes	47.22	48.56	47.36	63.70	2.91	0.118	0.165	0.016
Bacteroidetes	43.81	42.38	43.64	28.41	2.91	0.147	0.215	0.226
Proteobacteria	2.40	2.40	3.81	2.84	0.29	0.403	0.120	0.403
Actinobacteria	2.48	1.44	1.49	1.55	0.31	0.454	0.499	0.403
Deferribacteres	1.06	2.60	2.06	1.14	0.36	0.672	0.754	0.105
Synergistetes	1.06	0.86	0.59	0.99	0.09	0.568	0.370	0.123
Epsilonbacteraeota	1.58	0.90	0.34	0.33	0.17	0.252	0.006	0.270
Tenericutes	0.24	0.45	0.37	0.69	0.07	0.037	0.140	0.657
Cyanobacteria	0.09	0.11	0.22	0.13	0.05	0.703	0.292	0.611
Elusimicrobia	0.03	0.36	0.03	0.08	0.03	0.115	0.222	0.254
Others	0.08	0.15	0.11	0.17	0.08	0.098	0.561	0.874
Genera								
Bacteroides	20.38	17.05	28.68	18.02	2.30	0.135	0.315	0.425
Faecalibacterium	15.03	8.90	10.35	24.83	2.24	0.298	0.165	0.016
Rikenellaceae RC9 gut group	14.79	17.71	9.60	3.80	2.74	0.791	0.091	0.427
[Ruminococcus] torques group	4.12	10.74	8.41	4.71	1.06	0.457	0.658	0.014
Ruminococcaceae UCG-014	4.28	4.47	4.32	6.57	0.86	0.503	0.559	0.570
Desulfovibrio	2.00	2.07	3.38	2.20	0.29	0.334	0.193	0.280
Phascolarctobacterium	4.34	1.43	1.21	2.83	0.38	0.294	0.168	0.001
Megamonas	4.79	1.02	0.74	1.16	0.68	0.194	0.133	0.108
Alistipes	1.86	2.05	1.36	1.68	0.16	0.443	0.194	0.842
Mucispirillum	1.06	2.60	2.06	1.14	0.36	0.672	0.754	0.105
Others	27.35	31.98	29.90	33.05	1.48	0.214	0.557	0.809

## Data Availability

Raw reads of bacterial 16S rRNA gene sequencing are available in the NCBI Sequence Read Archive database (Accession Number: PRJNA906987). Other data that support the findings of this study were not deposited in an official repository, but they are available from the authors upon request.

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
