# Peer review of "Effects of Dietary Bacillus coagulans and Tributyrin on Growth Performance, Serum Antioxidants, Intestinal Morphology, and Cecal Microbiota of Growing Yellow-Feathered Broilers"

_animals, 2023, doi:10.3390/ani13223534_

Round 1

Reviewer 1 Report

Comments and Suggestions for Authors

Comments to the Authors of manuscript number: animals-2678175 entitled “Effects of dietary bacillus coagulans and tributyrin on growth performance, serum antioxidant, intestinal morphology, and cecal microbiota of growing yellow-feathered broilers.”.

In this study the effects of Bacillus coagulans and tributyrin supplementation on the health and growth of yellow-feathered broiler chickens was examined. The combination of these additives had synergistic effects, improving antioxidant capacity, intestinal morphology, and the composition of intestinal flora. This research offers valuable insights for developing effective and safe feed additives in the poultry industry.

1. Introduction:

There are various grammatical issues throughout the text, including run-on sentences, missing commas, and incorrect word usage.

Some sentences are overly complex, making it difficult to understand the intended message.

The same information is sometimes repeated within a short span, which can be confusing for the reader.

Some sentences are longer than necessary and could be condensed for clarity.

The text lacks clear paragraph breaks, which can make it appear dense and challenging to read.

L70 – what is stress in the case of bacteria?

L 64, 71, 68 – uniform the name

L84- reference needed

L 92-106  this part should be rephrased and some information should be shift to material and methods or discussed in the discussion part, and here should be clearly presented the hypothesis and goal of the study

2. material and methods: It should be explained why yellow-feathered broilers were chosen. Typically, broilers are kept until 42 day of their life.

Although, the study design is presented as a two doses, really there is only one dose. The study is poor.

What was the reason of the chose the dose studied? It should be explained.

L 129- why lighting lasts 24h?

L 146- what does it mean that “of average body weight” to what?

L 164- named these kits, and catalog number and minimal detection should be given

provide the specific PCR conditions, including the cycling parameters, to ensure the reproducibility of the results.

97% similarity threshold. While this is a common threshold, the rationale for choosing this threshold should be provided. Different studies may use different thresholds, and the reasoning for the specific choice should be explained.

3. Results: Figure 1 lacks the scalbars and the particular images seem to be at different magnification. It should be corrected

Author Response

请参阅附件。

Reviewer 2 Report

Comments and Suggestions for Authors

In this study, the author examined the effects of Bacillus coagulans and tributyrin supplementation on the growth performance, se-rum antioxidant contents, intestinal morphology, and caecal microbiota of yellow-feathered broiler chickens. This study is very interesting, and the experimental results are quite detailed. However, the manuscript readability could be much improved to better convey the importance of your study. With editing and some minor revisions, I feel that this manuscript will be suitable for publication.

1.      The Abstract part of the article has too many words and is not concise enough, please limit it to about 250 words.

2.      A total of 72 references were cited for this article. Nearly half of the references are from 5 years ago or even older. For innovative articles, references should be nearly 3-5 years. Please update references.

3.      Line 454, the author should modify the expression of B. coagulans count and conduct a full text review.

4.      Tables 5-7 belong to the same type of table, and it is recommended that the author merge the three tables into one table.

5.      The grammar of the entire article should be polished by someone whose native language is English

Comments on the Quality of English Language

 Moderate editing of English language required

Reviewer 3 Report

Comments and Suggestions for Authors

This study provides valuable insights into the effects of Bacillus coagulans and tributyrin supplementation on yellow-feathered broiler chickens. However, there are several points that could be critiqued for further clarity and consideration:

  1. Incorrect Formatting: The term "Bacillus coagulans" has not been consistently written correctly, and it should be italicized to indicate it's a scientific name. This oversight impacts the clarity and professionalism of the paper.
  2. Lack of References in Discussion: Numerous statements in the discussion lack proper referencing. This makes it challenging for readers to verify the accuracy of these statements or to delve deeper into the cited literature for a broader understanding of the context.
  3. Unclear ANOVA Test: The paper does not specify which specific ANOVA test was used in the statistical analysis. Providing this information is crucial for transparency and reproducibility of the study.
  4. Low Analyzed Crude Protein Level and Missing Units: The paper does not provide a rationale for the relatively low levels of analyzed crude protein. Additionally, no units are provided for the analyzed feed ingredients. This is crucial information for the readers to accurately interpret the results.

Addressing these points would enhance the clarity, accuracy, and professionalism of the paper.

Top of Form

Round 2

Reviewer 1 Report

Comments and Suggestions for Authors

i have no more comments